# A Novel Assessment of the Surface Heat Flux Role in Radon (Rn-222) Gas Flow within Subsurface Geological Porous Media

Ayelet Benkovitz [1], Hovav Zafrir [2,3] and Yuval Reuveni [1,4,5,*]

1. Department of Physics, Ariel University, Ariel 4070000, Israel; ayeletbe@ariel.ac.il
2. Geological Survey of Israel, Jerusalem 9692100, Israel; hzafrir@gmail.com
3. Faculty of Engineering, Bar-Ilan University, Ramat Gan 5290002, Israel
4. Eastern R&D Center, Ariel 4070000, Israel
5. School of Sustainability, Reichman University, Herzliya 4610101, Israel
* Correspondence: yuvalr@ariel.ac.il; Tel.: +972-52-5970648

**Abstract:** At present, Rn subsurface flow can be described only by diffusion and advection transportation models within porous media that currently exist. Even though the temperature is a strong driving force in climate and gas thermodynamics, the impact of the surface heating is missing from all gas flow models within geological porous media. In this work, it is shown that heating the ground surface by the sun, every day up to a maximum temperature at noon, creates a downward vertical temperature gradient related to the constant temperature in the upper shallow layer whose measured thickness is several meters. Undersurface, the Rn gas in the porous media is propelled in nonlinear dependency by the surface temperature gradient to flow downward, up to a measured depth of 100 m, revealing a daily periodicity with time delay depending on depth, similar to the diurnal cycle of the surface temperature. Moreover, regression analysis applied with the data implies a non-linear relationship between Rn and the temporal surface temperature. The relationship is non-linear and the best fit for it from a thermodynamic point of view is an exponential dependency. From now on, it will be possible according to the model to predict and extract, if required, by the time series of the surface-measured parameters (the ambient temperature and pressure), the semi-diurnal, diurnal, multiday, and seasonal Rn temporal variation at a shallow depth.

**Keywords:** radon thermal flow in geological porous media; radon exponential correlation to daily climatic surface temperature; radon cyclical signals undulate by ambient meteorological parameters





## 1. Introduction

Radon and its RDPs ([222]Rn decay products) are one of the main sources of near-surface gamma-rays, naturally produced from bearing uranium minerals, dust, and groundwater carrying weathered rock and dissolved Rn [1–8]. There are a few factors controlling Rn levels in the geological porous media, such as the various amounts of the nuclei parent in the soil, the concentration gradient of an existing Rn in the various rock layers that will govern diffusion, variable internal pressure levels producing an advection process from shallow inground towards the open air (process defined as exhalation), and water content of the media [2,9–11].

Ground-level gamma enhancements are also attributed to radionuclides ([222]Rn, and its RDPs) scavenged by raindrops or snow crystals [4,12–14], Thunderstorms Ground Enhancements (TGEs) associated with the acceleration of protons in strong electric fields within and below, secondary cosmic-ray flux, solar storms, and enhanced geomagnetic activity [15]. Further to fluctuating behavior due to regional events, sub-ground Rn regularly varies throughout seasons and along the day [10,16–19], implying a lithosphere–atmosphere coupling mechanism [20,21]. In addition, increased gas emanation through geological fault movement can result with enhanced Rn levels [20,22–24]. Thus, Rn is suggested to be an indicator of pre-tectonic events, such as earthquakes and volcanic eruptions [25–28], even

though there is not yet a clear model of this Rn–tectonic activity relationship that may allow predicting near-future seismic events [29–31]. Gamma and alpha radiation can be produced by other radioactive elements, but unlike Rn, they are nonmobile solids, and therefore, their decay products are in an equilibrium state with little variation if new material is added or removed in the environment. A significant change in radiation levels, however, can be attributed to a mobile particle such as Rn radioactive gas with a half-life of ~3.8 days, which enables the noble gas atom to move within the porous media under the influence of different driving forces. The change in the Rn signal can be described as a sum of changes attributed to regional phenomena as mentioned above, coupled with changing in climatic temperature and pressure [32–34]. The wind regime can also contribute to convection in the upper subsurface, and relative humidity has an impact on air density as well, but both to a lesser degree [21].

The movement of radon gas (Rn-222) in the geological media has occupied many researchers in the last fifty years [10,16,18,19,25,35–47]. The studies on the movement of radon gas in porous media within the subsurface succeeded in formulating movement equations for radon in air or water-pore-spaces that take into account two parameters, as driving forces: the Rn local concentration gradient and the local pressure gradient.

As for the present, current models concentrate on explaining subsurface gas flow by diffusion and advection processes derived from concentration and pressure gradients. The modified Darcy–Fick laws are the best model so far to describe Rn in porous media [35–37]:

$$\beta \frac{\partial C_a}{\partial t} = \nabla \cdot (D \nabla C_a) + \frac{K}{\mu} \cdot \nabla P \cdot \nabla C_a - \beta \lambda C_a + S \tag{1}$$

where β is the partition-corrected porosity. This was suggested by Andersen [38], which takes into account the partition between the air and water phase; Ca is Rn concentration in the air phase (Bq m$^{-3}$). There are four components on the right-hand side of Equation (1). Describing Rn flow:

1.　$\nabla \cdot (D \nabla C_a)$—diffusion transport. D is the Rn diffusion coefficient in air-filled pores (m$^2 \cdot$s$^{-1}$); $\nabla$Ca is Rn concentration gradient (Bq·m$^{-4}$).
2.　$\frac{K}{\mu} \cdot \nabla P \cdot \nabla C_a$—advective transport. K is intrinsic permeability (m$^2$); μ is the dynamic viscosity of air (1.83·10$^{-5}$ kg·m$^{-1}$·s$^{-1}$ or Pa·s at T = 293 K).
3.　$\beta \lambda C_a$–Rn decay. β is the partition-corrected porosity; λ is the Rn decay constant (2.1 × 10$^{-6}$ s$^{-1}$).
4.　$S$–Rn production rate. $S = \eta \rho_b \lambda C_{Ra}$ (Bq m$^{-3} \cdot$s$^{-1}$). η is the sum of the emanation fractions to the different phases (air, water and adsorbed); $\rho_b$ is bulk density (kg·m$^3$); $C_{Ra}$ is the concentration of immediate Rn parent $^{226}$Ra [36,48,49].

However, even though the existing model proved to be quite satisfactory [48,49] for non-isothermal Rn transport, the physical equations do not include ground surface temperature, which is one of the most significant driving forces in gas thermodynamics within the ground [50]. Furthermore, in order to be able to use Rn as a natural hazard precursor, it is essential first to understand its basic temporal periodical behavior. A high-resolution study of Rn behavior under natural conditions, coupled with ambient surface atmospheric data, may shed further light on the dependencies that exist between undersurface shallow gas flow and surface climate affection—mainly ambient temperature and pressure. In addition, this study's findings can be reflected in the study of other subsurface gases such as $CO_2$.

Here, we present comprehensive results from long-term Rn monitoring in depth, during 2015–2018, at high temporal resolution, along with ambient surface temperature and barometric pressure measurements. An experimental system was set up in the Northern part of Israel (Figure 1); three Rn gamma-ray detectors were inserted into the airspace of an abandoned borehole, which was built as an iron pipe 20″ radius, which descends sealed to 80 m depth and has a perforated extension of more than 30 m.; two Rn gamma detectors were set up at depths of 10 and 60 m, the third under the water table at 88 m, and

an additional Rn alpha detector at 40 m depth [21,30,31,51]. Inclusively, it was proven in our previous studies that the Rn gamma detectors at depth monitor the gas flow within the surrounding bedrock [52]. Semi-diurnal, diurnal, multiday, and seasonal Rn temporal variation, and their relationship to surface temperature, were studied. Daily profiles of Rn showed an exponential correlation to daily surface temperature, with a lagging response between the Rn gamma detectors according to the detector location at depth [30]. In addition, it was revealed that there is a significant difference in Rn daily profiles, divided according to season. In conclusion and comprehensively, regression analysis applied to the data implied a non-linear relationship between Rn and the temporal surface temperature. The best fit for it from a thermodynamic point of view is an exponential dependency.

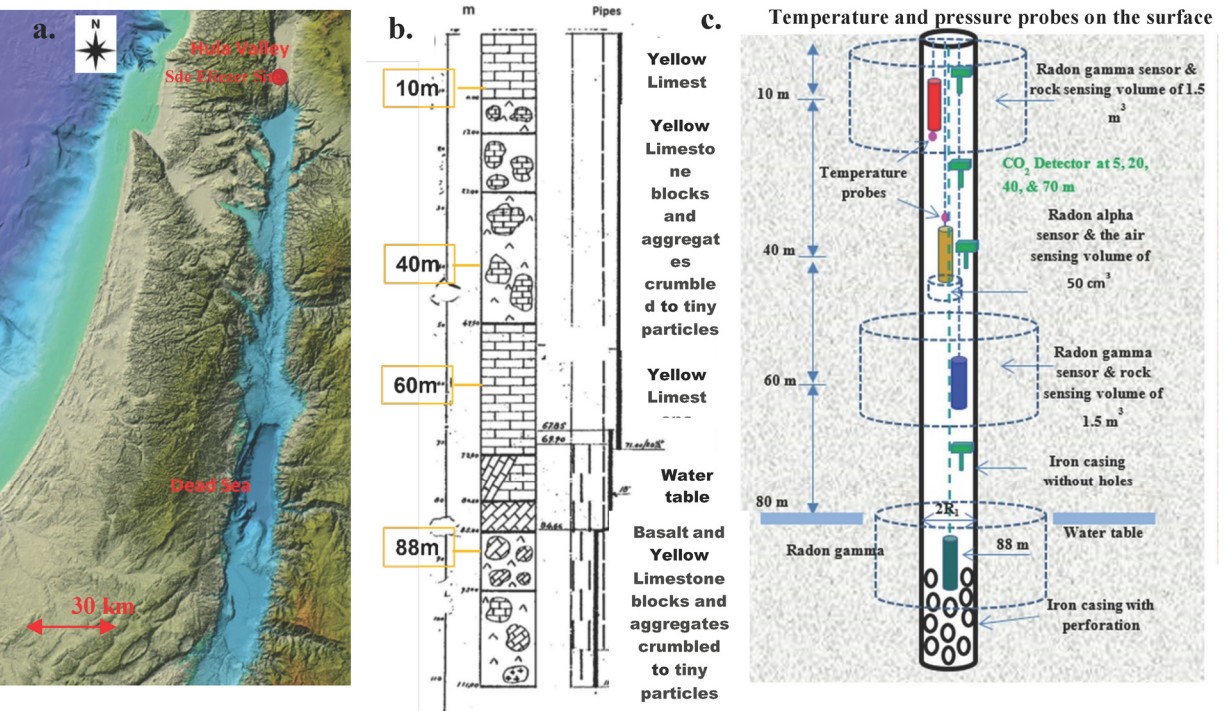

**Figure 1.** (**a**) Sde-Eliezer located in Northern Israel on the western border of the Hula Valley. (**b**) Vertical profile of the borehole rock composition with orange boxes which note at what depth a detector was positioned. The log is characterized by limestone. The soil is organic-rich consisting of $^{238}$U-organic complexes, which contribute to the generation of Rn [30]. (**c**) Schematic illustration of the experiment setup. The dashed cylinders represent the detecting volume of each sensor. Detector at 88m depth is under the water table and has a smaller detecting volume as the gamma radiation has a higher absorption rate in water than in rock [53].

## 2. Experimental Setup

The experimental system was set up in a 1 m diameter abandoned water-well with iron-pipe casing, in Sde-Eliezer, near the Hula Valley (33.050616°N 35.559039°E, 159 m above sea level) (Figure 1). The country limestone rock column under the surface of the Hula valley consists of organic substances (peat and lignite) which contain U-organic complexes [31,54] and contribute to the gamma-ray background within the Rn detectors. As the rock column is not entirely homogenous, this will contribute to the different backgrounds of the Rn measured. Radon was monitored at four different depths using the following detectors:

At depths 10 and 60 m: Rn detection by gamma-ray measurements by BGO (1.5″ diameter × 3″) crystal scintillators, equipped with an electronic total count Single Channel Analyzer (Type 36 B 76/1 M-HV–E3 –X2 of Scionix, La Bunnik, The Netherlands). The sensitivity of the detector is 1.8 counts per 15 min are equivalent to concentration of 1 Bq/m$^3$ [55]. At the depth of 88 m: Rn detection by gamma measurements by NaI (Tl)

(1.5″ diameter × 3″) crystal scintillator (of Scionix, La Bunnik, Holland). The sensitivity of the detector is 2.4 counts per 15 min are equivalent to concentration of 1 Bq/m$^3$.

At depth 40 m: Rn detection by alpha measurements by Barasol BT45N (Algade Inc., Bessines-sur-Gartempe, France). The sensitivity of the detector is 0.08 counts per 15 min, equivalent to a concentration of 1 Bq/m$^3$. The alpha detector measures Rn within the borehole airspace that enters by diffusion to the detector's sensing volume of 50 cm$^3$. Therefore, alpha reflects the temporal borehole airspace Rn concentration.

Gamma detectors, on the other hand, can detect gamma rays up 35 cm from the bedrock surrounding the borehole's iron pipe and have a view angle of 45° degrees, creating a total bedrock sensing volume of 1.5 m$^3$ that contributes to the detectors' readings [30]. The systematic measuring error of detectors is defined as e = $\sqrt{n}/n$, where n is the number of counts during the measurement time interval of each detector. According to the different counting rates of both radiation detectors and because of the low efficiency of the alpha detector, gamma and alpha detectors have an error of ~3% and ~20%, respectively. In order to have a higher accuracy by the alpha detector, it is essential to increase the counting time interval to 8 h instead of 15 min. This diversity in accuracy is due to the large difference of 3 orders of magnitude between the high sensitivity of the gamma detector versus the alpha-based detection technology [55].

During the entire period, surface atmospheric temperature and barometric pressure were measured using the CR1000 of Campbell Scientific data logger (CR1000, Campbell Scientific, UT, USA). The temperature was measured by a thermocouple probe (type T, Omega Engineering, Manchester, UK) and pressure by a barometric pressure sensor (CS106, Vaisala, Vantaa, Finland). A Schematic illustration of the experimental setup is found in Figure 1c. Data for the years 2015–2017 were recorded in 15 min resolution time intervals. Data for the year 2018 were taken every 30 s, and therefore, surface temperature and pressure averages were calculated, and gamma and alpha data were summed, both in 15 min time windows. Time-series analysis, statistical and regression methods were used for analyzing the data. The analysis was conducted under a Python environment using built-in and customized functions built upon libraries Datetime, NumPy, Pandas, and SciPy. The libraries Matplotlib and Seaborn were used for visualization of the results.

## 3. Results

Figure 2 presents ~4 years (16 January 2015–5 October 2018) of Rn measurements at 10, 40, 60, and 88 m depth (in counts per time interval), and measurements of surface atmospheric temperature, and surface barometric pressure, in high-resolution time intervals of 15 min. Between the dates 4 September 2017–14 March 2018, there was equipment failure, and no measurements were acquired. The Rn gamma-ray detector at 88 m depth was added in November 2015. The low readings up to July 2016, and thereafter December 2016, are attributed to water covering this detector and absorbing gamma-rays originating from the rock column; during the summer of 2016, the water level receded, increasing the measurement rate of the low readings up to July 2016, and thereafter December 2016, which are attributed to water covering this detector and absorbing gamma-rays originating from the rock column; during the summer of 2016, the water level receded, increasing the measurement rate of the detection of Rn from its bedrock surroundings, as the upper Rn gamma sensors. A close-up of five days of summer (1–5 August 2016, solid line) and winter (25–29 December 2016) were plotted together in Figure 3. All radioactive datasets show a clear annual periodicity correlating positively with the surface temperature and negatively with the surface barometric pressure. Conversion units between counts to Becquerel/m$^3$ are gamma sensors at depths of 10 and 60 m −1.8 counts per 15 min are equivalent to the concentration of 1 Bq/m$^3$; gamma sensors at depth 88 m −2.4 counts per 15 min are equivalent to the concentration of 1 Bq/m$^3$; alpha particle detector −0.08 counts per 15 min are equivalent also to 1 Bq/m$^3$. Radon counts acquired by the detectors were converted to Becquerel/m$^3$ units (Bq/m$^3$) according to a comparative study between different Rn gas detectors carried out in the past, and as specified in the experimental setup paragraph [55].

As mentioned earlier in the introduction, there are other non-mobile elements inside the rock that go through radioactive decay and produce background gamma-rays. On the other hand, the alpha background can drop to zero, indicating that there are times where there is no radioactive element streaming within the pipe airspace.

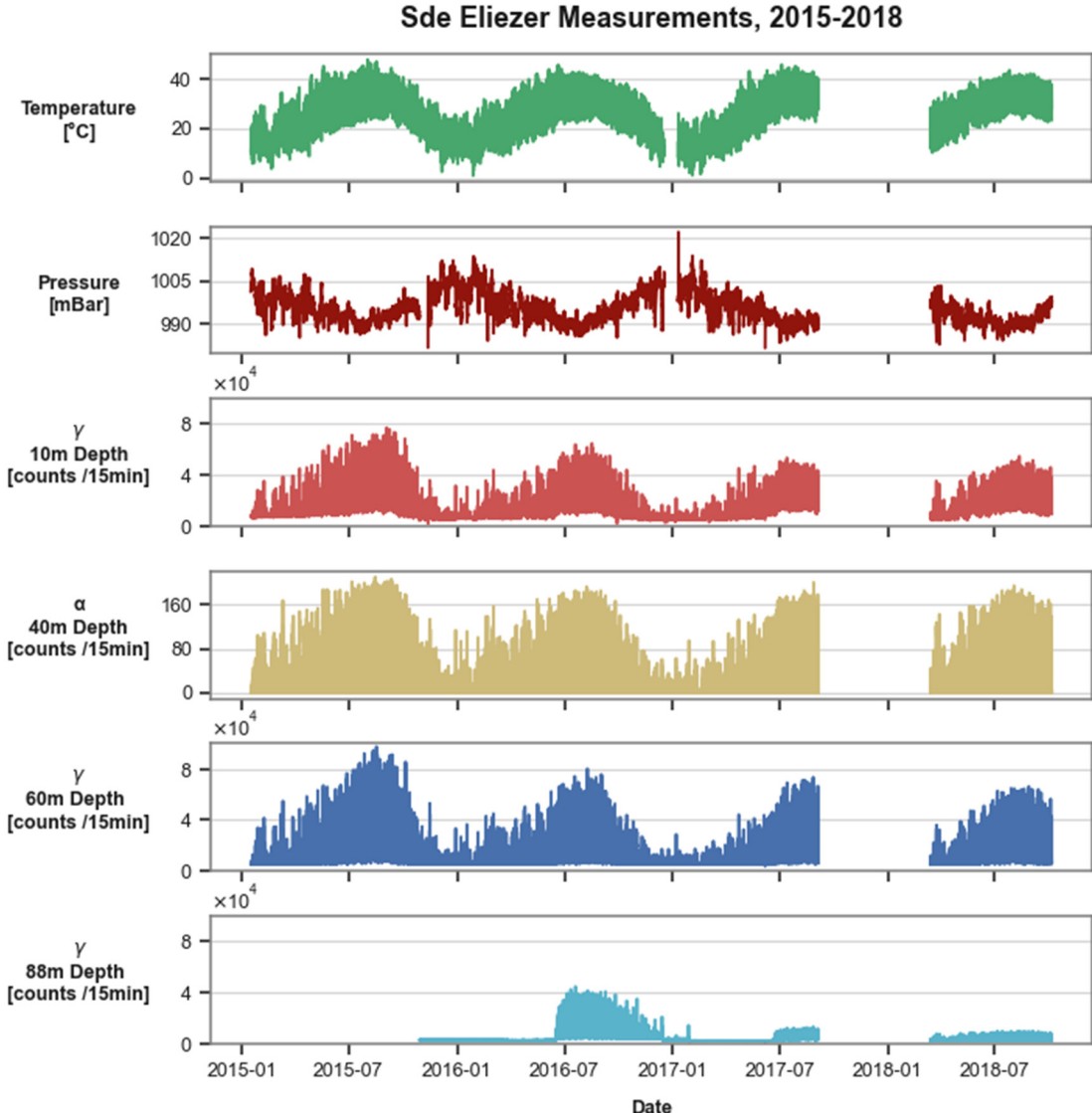

**Figure 2.** The 2015–2018 Sde-Eliezer measurements of Rn counts at 10, 40, 60, and 88 meters depth, and surface temperature, and barometric pressure, at 15 min resolution time.

In order to analyze the temporal variation of the Rn gas concentration within the bedrock porous media only, the gamma background was subtracted from the gamma detectors' raw data at 10, 60, and 88 m depth. It is important to reemphasize that this method allows monitoring the change in flowing Rn, which leaves the solid grains of the rock column that surrounds the pipe. The Rn that has not emanated (has not exited) its mother grain is part of the donors' constant background gamma-ray signal and is subtracted with the other contribution of the solid non-mobile radioactive elements. This background varies slightly with the season (as can be noticed in Rn-10 m in Figure 2) and therefore, each day was subtracted by an average of its lower values (quantile 0.02). Regarding the alpha detector data, when there is no barometric pumping inside the borehole, Rn stops streaming inside the pipe and alpha detector acquires zero values. In winter, the surface cold air during the night and day, and the high level of winter barometric pressure, inhibits

barometric pumping inside the borehole and are also responsible for reducing in general the concentration of radon gas that is free to move within the geological porous media by a factor between 3 to 7. This is also clear from the horizontal pink line in the temperature plot, representing the temperature at 10 m depth. Elevation in nocturnal Rn was possible at times when the surface temperature was greater than the temperature existing in depth. Hence, the lack of seasonal nocturnal Rn signals between November to March, in Hula Vally, Israel.

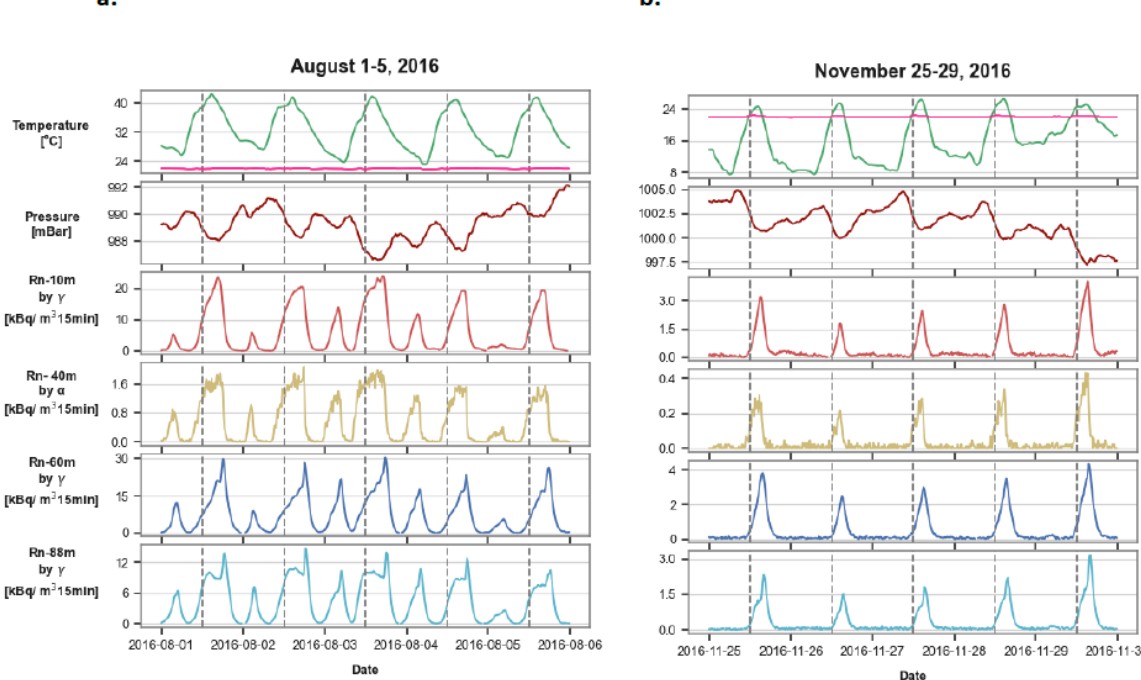

**Figure 3.** Five days between (**a**) 1–5 August 2016 and (**b**) 25–29 December 2016, representing summer and winter seasons, respectively. Dashed vertical lines indicate noon time. Note that the scale between the two figures is different; August has significantly higher Rn readings than November. Additionally, during August there is a distinctive Rn peak in all depths during the night that is missing from November days.

### 3.1. Daily Profiles

Daily profiles of surface temperature and barometric pressure, and Rn values at 10, 40, 60, and 88 m depth according to seasons, are presented in Figure 4a. The profiles consist of all measurement data points according to the number of days sampled. Magnitude-wise, all variables differ significantly between the two seasons. The local summer (months May–October) and winter (months November–April) were determined according to surface temperature during those months at the experiment site, and not by the traditional 3 months season division (Figure 4b). Daily maximum surface temperatures were averaged according to month and day within the month. T = 30 °C was chosen since days with lower surface temperatures appear to fluctuate more, a behavior characteristic of wintertime.

Radon shows two daily peaks: a strong one during the daytime and a weaker one during the night.

a.

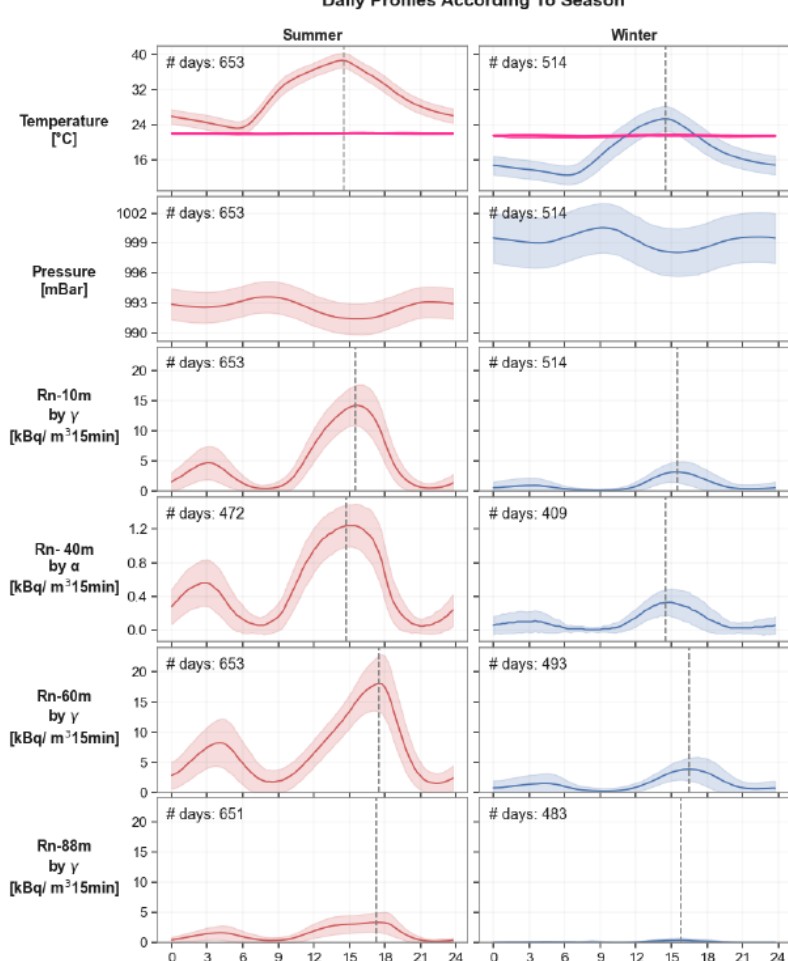

b.

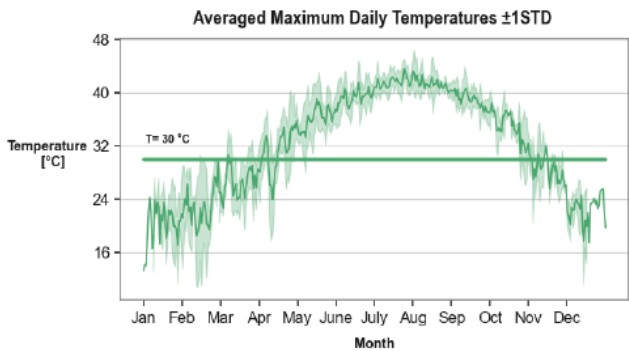

**Figure 4.** (**a**) Daily profiles ($\pm$0.5 STD) of surface temperature, barometric pressure, and Rn at 10, 40, 60, and 88 m depth. The pink line in the temperatures plot represents the daily profile of temperature at 10 m depth. Dashed vertical lines represent the daily time of maximum temperature and Rn. The figure shows a clear difference in scale between the two seasons. Radon profiles show two peaks per day, correlating to the minimum in surface barometric pressure, which has a semi-diurnal pattern. The daytime peak is significantly higher than the nocturnal peak, implying for the main driving force effecting Rn behavior—the surface temperature. There is a lag in the response of the Rn signals with depth, as reflected by 60 and 88 m peaks shifted to the right relative to the 10 m one. The generally low winter readings are attributed to low temperatures and weak gradients, which are not sufficient to force subsurface Rn flow. (**b**) Averaged daily maximum of surface temperatures according to months and days. Months with T > 30 °C are summer, and those below—winter.

Levintal [21] established that barometric pumping controls the movement of air inside the borehole, with an average velocity of 3 m/min. The iron pipe inside the borehole is wall-sealed and therefore, Rn reaching the alpha detector at 40 m depth is assumed to come only from the surrounding rock, via the bottom opening where the pipe-iron cast ends, and if water exists, from soluble Rn. In the same process, the $CO_2$ also flows into the bottom of the pipe [21,31]. As the temperature at 10 m depth is already constant, there is no temperature gradient that contributes to air convection and the local barometric pressure gradient is the sole reason for air movement in the borehole in those depths. At night, the nocturnal semidiurnal (12 h) pressure cycle (created by barometric pumping inside the borehole), carries Rn from the depth to the surface. Most gamma readings in summer night's signals (Figures 3a and 4a) are due to the Rn traveling along the borehole, accompanied by a Rn weak contribution from the gas variation within the bedrock, due to a slightly reduction in the ambient surface temperature gradient leading to vertical flow and advection.

During the daytime, however, the sensors detect twice as much Rn, indicating the affection of the additional driving force at play, i.e., surface temperature, with a diurnal (24 h) periodicity. During the daytime, Rn has a positive correlation with this variating temperature. It is also noticeable that there is a lag in Rn profiles with depth, suggesting a delayed response of Rn flow downward within the bedrock because of the heating of the surface. The lag depends on the downward velocity of the Rn movement in the surrounding bedrock [30,51,52]. Another study [17] also showed similar results from Rn monitoring at 10 m depth. The low surface temperatures during winter do not generate a strong enough downward temperature gradient and hence the much lower Rn daily signals during that season (Figures 3b and 4a). This difference between summer and winter for single days can also be found in Figure 4. In addition to the difference in scale, the nocturnal Rn-peak is almost non-existent during the winter as the surface cold air during the night, and the winter high surface barometric pressure inhibits barometric pumping inside the borehole (Figures 3 and 4a).

The drop in the Rn level's intensities between 60 and 88 m in Rn daily profiles (Figure 4a) may be due to the changes in the type of the borehole's surrounding bedrock at 85 m from limestone to rock consisting of both basalt and limestone (Figure 1b). Another possibility is the weakening of the Rn flow under the influence of the surface temperature impact with depth. The main assumption is that the heavy radon gas atoms (five times heavier than most of the air components in the porous media) during a mostly inelastic collision, transfer within the porous media to more and more identical atoms, their energy and the direction of their downward movement. Energy losses increase with depth and limit the range. In addition, the phenomenon fades with seasonal cooling, and the strength of the signals decreases by a factor of three to ten times, between the summer peak and the winter peak. The fact that the daytime peak at 88 m is wider than in other depths can suggest that internal processes within the porous media such as diffusion and advection are more pronounced because of the variation of the internal properties of the country rock's porous media, but further research is needed in order to determine that.

### 3.2. Temperature on the Surface and Rn within the Ground Relationship

Daily maximum values of surface temperature and Rn were chosen to explore the overall impact of ambient surface temperature on underground Rn for inter-day resolution.

To study the relationship between these two variables, Rn signals at depths 10, 40, and 60 m (Rn-88 m samples were limited and therefore not included) were first filtered according to the daily profile, and the following days were excluded:

- Days with the low signal where there were no detectable peaks (daily maximum Rn < twice the largest minimum value of the entire dataset. The values are 0.356, 0 and 0.396 kBq/m$^3$ for Rn at 10, 40, and 60 m depth, respectively).
- Days where daily maximum Rn appeared ±3 h from the average time of maximum Rn for that specific month.

The profile in which the two daily successive signals (noon and nocturnal signals) expanded to a wide one that lasted more than 12 h. This behavior is assumed to be related to geodynamic activity [30]. Figure 5 shows an example of Rn at 60 m depth with a wide signal considered to be related to pre-seismic tectonic event of a 5.5-magnitude earthquake originated in the Gulf of Eilat, a segment of the Dead Sea fault zone, near Nuweiba. The number of excluded days appears in Table 1.

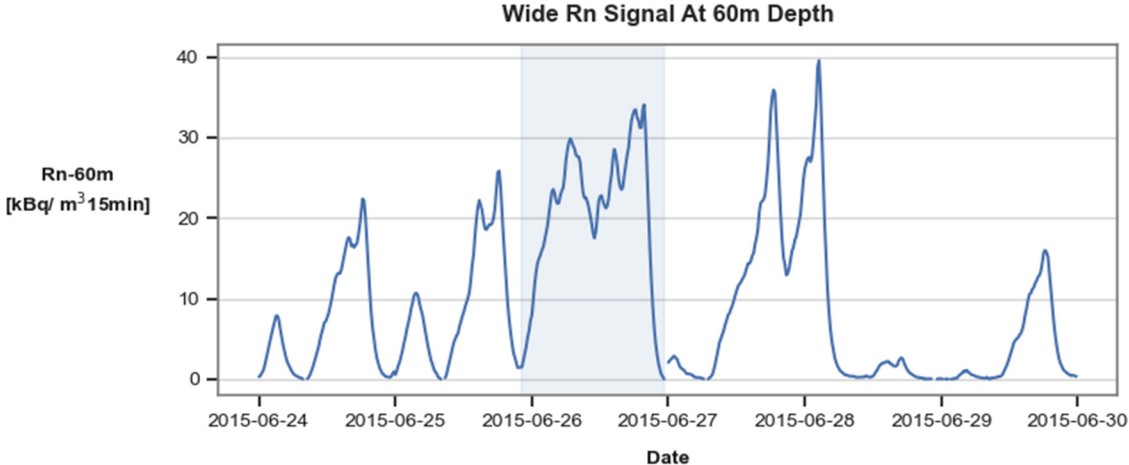

**Figure 5.** Radon signal at 60 m depth between 24 June 2015 to 29 June 2015. The shaded Rn signal of June 26 is particularly wide and lasted 23 h (shaded area between 25 June 22:00–26 June 23:00). This signal appeared a day before a 5.5-magnitude earthquake originating near Nuweiba, on one of the segments of the Dead Sea Transform. Occasionally, there are traces of these geodynamic events in Rn signal days after as can be seen from the non-periodical behavior of Rn on 27–28 June.

**Table 1.** Number of days before and after filtering the data.

| Depth [m] | Before Filtering Data | After Filtering Data |
|:---:|:---:|:---:|
| 10 | 1167 | 999 |
| 40 | 1167 | 995 |
| 60 | 1167 | 991 |

*3.3. The Temperature Dependency of the Rn Flow within the Borehole Airspace and in the Bedrock*

As mentioned before, in the current system, the flowing Rn inside the borehole contributes an additional quantity to the Rn measured by the gamma detectors and adds up to Rn detected from the surrounding rock. This implies that gamma sensors detect two different Rn flow contributions simultaneously. In order to study the impact of the surface temperature only, one needs to take into consideration this low contribution of the Rn atoms streaming up under the gas barometric pumping within the borehole airspace and subtract it.

The daily time intervals in which the maximum Rn flow is detected at 10 m and 40 m depths, and overlap on many occasions, imply an additional contribution of the streaming Rn inside the borehole to the counting of the gamma detector at 10 m depth. Nonetheless, Rn data from these two detectors show a different relationship to the surface atmospheric temperature, implying different processes involved in the Rn flow mechanism inside the borehole and within the rock column (Figure 6). There are different processes involved in Rn flow within the rock column and inside the borehole. During the daytime, the air column above and within the borehole expands vertically under the heating of the sun, creating a negative pressure gradient and inducing barometric pumping inside the borehole, carrying Rn from the bottom of the pipe into the open atmosphere. In the bedrock surrounding the sealed borehole iron pipe, Rn is flowing downward. In general, it can

be seen that the Rn concentration measured in the borehole, at any ambient temperature, by the Alpha detector at 40 m depth is about one-tenth of the general Rn concentration measured by the gamma detector at 10 m within the surrounding rock. Because of the geometry, the viewing angle of the gamma detector into the deep borehole is very narrow and therefore the contribution from the radon flowing in the pipe relative to the radon measured at the same time beyond the borehole pipe is even lower than one-tenth.

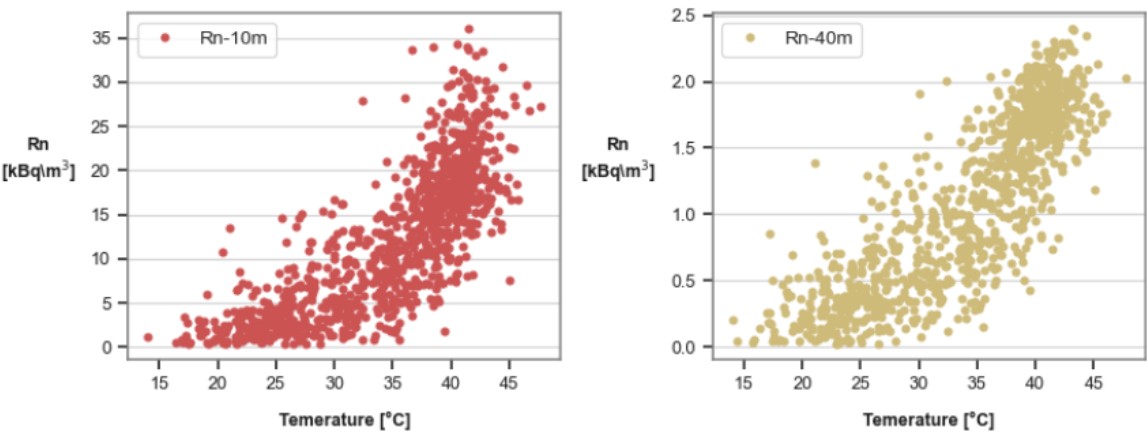

**Figure 6.** Relationship of maximum daily Rn at 10 m detected by gamma detector (**left**) and Rn at 40 m detected by alpha detector (**right**) against surface temperature. The gamma detector (Figure 1c) measures Rn from the surrounding country rock, whereas the Alpha detector measures only Rn streaming inside the borehole. Even though the maximum Rn measured by these two detectors appeared around the same daily time, the plots show different relationships with the surface temperature; Rn within rock at10 m depth inclines to exponential, while Rn at the borehole airspace at 40 m depth creates a linear trend.

Radon at 60 m depth reaches its maximum value later in the day (Figure 4a), with a lower contribution from Rn streaming up inside the borehole (see vertical lines of temperature and Rn-60 m). To further study the relationship between the surface atmospheric temperature and Rn flowing in the porous media, regression was applied to daily maximum values of Rn recorded at 60 m depth by the gamma-ray detector after simultaneous subtraction of the Rn measurement at 40 m, taken at the same time by the Rn alpha detector. For repeatability testing, regression was applied to four different time intervals separately (Figure 7) (e.g., 1 October 2016–30 September 2017). The weather during October is stable and mild and therefore it was a logical point in time to divide the data accordingly. Figure 7a presents the relationship between Rn levels at 60 m depth and surface atmospheric temperature in accordance with the four time periods, each with its fitting equation. Each point in the plots represent a maximum value of Rn against the maximum surface temperature per the same day. To better observe the seasonal variation of Rn, winter and summer data points were, respectively, plotted with hollow and full circles. An additional plot (Figure 7b) shows all four regression lines together, where the black line represents the overall model for Rn at 60 m depth. In all plots in Figure 7, Rn shows an exponential relationship with surface atmospheric temperature in the form of

$$a \times e^{b \cdot T} \qquad (2)$$

where *a* is the estimated initial Rn concentration at a minimal temperature, T, of the specific time period, and *b* is the annual natural growth rate in Rn levels. The exponential relationship indicates a non-linear relationship between the subsurface gas flow and surface atmospheric temperature. In other words, the amount of Rn at a depth of 60 m is exponentially dependent on the surface heating temperature above the ground; Rn rises

exponentially with rising surface air temperature. The partial graph of 2018 in Figure 7a 'March–September 2018' is due to the lack of winter samples between 1 October 2017–13 March 2018.

It is essential to restate that the subground temperature in the shallow depth of a few meters (10 m in our case) is already constant [21,30,31] (Figures 3 and 4a), and the energy that induces Rn movement in the rock column comes from the temperature on the surface, which heats the surface layer of the ground like an inverted pan. Radon molecules in the shallow surface are energized by the heat of the sun and forced to flow downwards, escaping the heat source above [51,52]. The delayed response of Rn with depth reflects the time it takes for Rn to travel downwards as a result of the kinetic energy that was given by the external surface temperature. As seen from the plots, parameters *a* and *b* differ slightly between the years. There are several factors that influence these parameters controlling Rn flow in a rock column, like porosity and grain size of the geological porous media, along with more temporal factors such as water content within the porous media as a result of the annual rainfall. All these factors together are enfolded in parameters *a* and *b*. Nevertheless, data in all four plots exhibit an exponential relationship, and all regression lines fall in, or in approximation to, the bold black line in Figure 7b, which represents the overall exponential equation of the entire dataset ±1 std (in blue shaded area).

To test whether the contribution of Rn traveling inside the borehole (measured by the alpha detector at 40 m depth) affects the model, one period of time (1 October 2016– 30 December 2017) was tested (Figure 8); regression was applied to maximum daily Rn recorded at 60 m depth, without subtracting it with Rn at 40 m. To check the goodness of the exponential fitting, the model was run 100 times, both for entire datasets (including all years) of maximum Rn at 60 m subtracted by Rn at 40 m and Rn at 60 m and Rn at 60 m without subtraction. In each run, the data were divided randomly into 80% as a train dataset and the other 20% as a test set. Table 2 presents the Coefficient of Variation percentage (%CV), Sum of Squares Error (SSE) and Root Mean Square Errors (RMSE) for the parameters *a* and *b*. CV is defined as the standard deviation divided by the mean of the dataset (STD/Mean), which gives an indication of how much a value is dispersed around its mean. SSE is ($\Sigma((y - \hat{y})^2)$), and RMSE ($\sqrt{\frac{SSE}{n}}$) is the average difference between the actual data to the model given in kBq/m$^3$. The low percentage CV of SSE received from 100 executions of the exponential model in both cases implies the high precision of the model. If otherwise, dispersion would have been greater and reflected by higher %CV values. As RMSE fraction and %CV SSE are similar in both cases, and the %CV of parameters *a* and *b* are both low as well, there is no inclination that streaming Rn inside the borehole disrupts the observation of Rn flow within the rock column. The visual evidence from the figures enforced by the statistical tests brought in Table 2 implies that the exponential model describes in close approximation the processes of Rn flow in porous media as a function of surface atmospheric temperature in annual resolution. Table 2 statistically verifies the existence of the exponential relationship between Rn and temperature to test how close the individual datapoints are to the overall model, and two months, February and June 2016, were plotted along the predicted Rn values and their errors were calculated (Figure 9). The data points in Figure 9 fall around the predicted Rn values. The calculated RMSE are 3.639 and 6.414 kBq/m$^3$ for the months of February and June 2016, respectively. The error of June is greater than February's error as there is less movement of Rn in cold temperatures (as seen in the daily profiles), but for both months this error is ~24% from the range of values of each month (RMSE/Rn$_{max}$-Rn$_{min}$). The error represents the portion of Rn flow related to driving forces other than temperature, while the other ~76% can be attributed to the temperature effect on Rn flow in the annual resolution.

**a.**

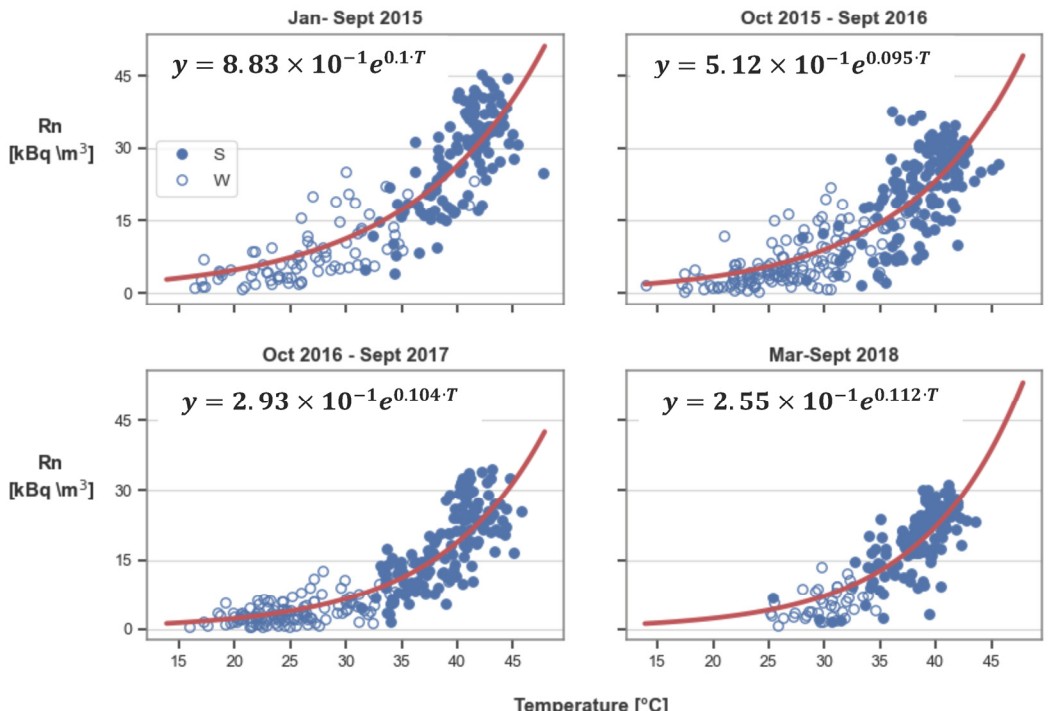

**b.**

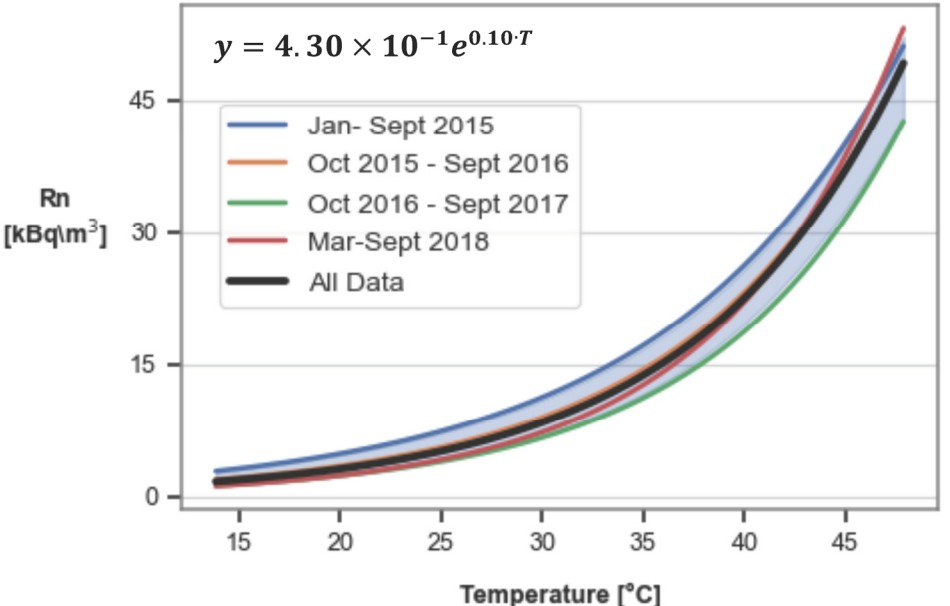

**Figure 7.** (**a**) Relationship between maximum daily Rn at depth of 60 m after subtracting contribution of Rn streaming inside the borehole, to atmospheric surface temperature, with fitting equations and according to the various time periods. Winter and summer datapoints are noted by hollow and full circles, respectively. All regression lines imply for an exponential relationship between Rn in depth to surface temperature. The lack in Rn samples during 2018 might be the cause for the less pronounced relationship that the data points create in the plot. (**b**) All four regression lines with overall model in (in bold black line) with ±1 STD. (**b**) The bold black line represents the overall exponential equation of the entire dataset ± 1 std (in blue shaded area).

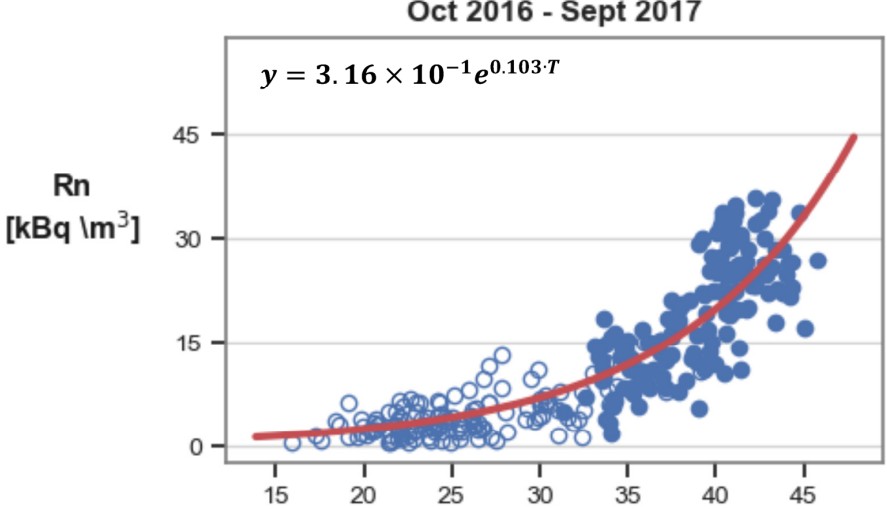

**Figure 8.** Regression applied to maximum daily Rn at 60 m depth, without taking into consideration the contribution of Rn streaming in the borehole (i.e., Rn at 40 m that was measured by the Alpha detector was not subtracted from Rn at 60 m), against atmospheric temperature. The exponential relationship is still strong after not considering Rn flow within the borehole, which is derived by pressure pumping (see the upper right plot in Figure 7a for regression of the same time period for data subtracted by Rn at 40 m).

**Table 2.** Goodness-of-fit.

|  | Maximum Rn-60 m $-$ Rn-40 m | Maximum Rn-60 m |
|---|---|---|
| Parameter *a* [% CV] | 6.13 | 5.68 |
| Parameter *b* [% CV] | 1.7 | 1.53 |
| SSE [%CV] | 11.35 | 11.26 |
| averaged RMSE [kBq/m$^3$ ] | 4.82 $\pm$ 0.28 | 5.15 $\pm$ 0.29 |

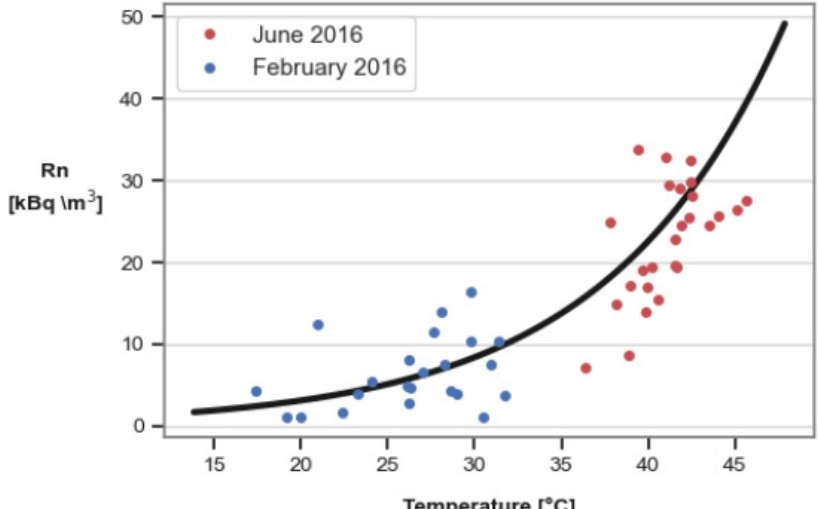

**Figure 9.** Maximum Rn at 60 m of months February (in blue) and June (in red) 2016 plotted together with the predicted Rn values (black line) according to the overall model in Figure 7b. The datapoints concentrate around the predicted Rn values, affirming the model.

## 4. Discussion and Conclusions

Until now, the flow of subsurface gases has been described as a function of advection and diffusion. In this article, we present for the first-time relationship between deep subsurface Rn and its dependency upon the surface atmospheric temperature spanning ~4 years of high-resolution measurements. At the same time as carrying out this research work, in a laboratory study, it was shown that heating the surface flows the radon downward into the porous medium, while cooling the surface relative to the ground causes the exhalation of radon through the surface into the atmosphere. When the heating and cooling cycle of the surface was changed from a one-day frequency as in nature to heating in cycles of four, six, and eight days, a cyclical flow was measured within the container of the porous material at the same frequencies, respectively. When the surface temperature was kept constant, no upward or downward flow of radon was observed at all [50,51].

The formal expression presented for the dependence between the flow of radon gas in the subsoil, and the surface temperature that produces the resulting thermal gradient in relation to the ground, will serve in the creation of an improved model for the Darcy–Fick equation. It can be estimated that it will be accomplished by eventually determining an additional function F(Ca, $\Delta$T, $\kappa$, Ek, T, t) to effectively describe and quantify the radon thermal flux from the surface heated by the sun to the subsurface porous rock and vice versa. It will allow us to determine the gas flux, its direction, and range, as a function of Ca—the concentration of radon in the geological porous media airspaces, $\Delta$T—the temperature gradient in the upper porous layer, $\kappa$—the thermal conductivity of the gas in the porous media, $E_k$—the transitional kinetic energy of the radon atoms as a function of the temperature T ($E_k = 3/2kBT$), and t—the time.

Although the shallow ground temperature is constant in the first few upper meters, the impact of the surface temperature downward gradient drives Rn gas down, up to tens of meters in the country rock column [10,21,30,46]. Daily profiles showed a significant difference between Rn daytime and nocturnal peaks, suggesting an additional driving force inducing this diurnal peak. It was suggested previously [18,30,31,50,51] that the surface temperature gradient forces the near-surface Rn to flow inward into the rock, escaping the heat source.

Based on the capabilities of the Rn gamma-ray detectors and the Rn alpha detectors, to measure Rn in different natural environments such as solid geological media and in airspace, respectively, it became possible to monitor simultaneous two different modes of Rn temporal flow variations occurring in depth: Rn streaming inside open boreholes, caves, mines or tunnels induced by barometric pumping, and Rn movement in the bedrock porous media under the impact of surface temperature. A regression test of the dependence between the temporal variations of Rn content in the depth of the geological porous media, to the climatic ambient surface temperature during four years, 2015–2018, revealed a clear exponential, non-linear, relationship in the form of *a*\*exp (*b*\* T). Testing the overall model with months of February and June 2016 databases implied that the surface temperature effect has a significant weight in controlling Rn semi-daily, daily, seasonal, and annual behavior. And on the other hand, Rn streaming inside an open space as a borehole, for example, did not show to affect this exponential model, which describes Rn flow within the rock column. Exponential equation parameters, *a* and *b*, can vary slightly with changing environments. Future research may test and determine these exponential parameters and their relation to environmental conditions such as porosity, grain composition, and water content of the porous media.

Nowadays, subsurface Rn transport is described mainly as a function of pressure and diffusion. We proved here that the main force that may drive gas in the porous medium in the subsurface, is the surface temperature gradient. And from this, it can be concluded that every geothermal source in the subsoil pushes natural gases to the surface through the rocks and not only through cracks or geological fractures and faults. In addition, as Rn is suggested as a precursor for earthquakes, understanding this gas basic periodical behavior in geological porous media is necessary in order to resolve cyclic signals of Rn in

the ground depth from random Rn signals that may be generated due to tectonic activity that precedes the appearance of earthquakes or volcanic eruptions.

**Author Contributions:** Conceptualization, H.Z. and Y.R.; Methodology, A.B., H.Z. and Y.R.; Validation, Y.R.; Formal analysis, A.B.; Resources, A.B.; Writing—original draft, A.B.; Writing—review & editing, H.Z. and Y.R.; Supervision, Y.R. All authors have read and agreed to the published version of the manuscript.

**Funding:** This research was funded by The Israeli Ministry of Energy, grant number 221-17-018.

**Data Availability Statement:** The data that support the findings of this study are available from the corresponding author, Y.R., upon reasonable request.

**Conflicts of Interest:** The authors declare no conflict of interest.

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
