# Peer review of "A Novel Assessment of the Surface Heat Flux Role in Radon (Rn-222) Gas Flow within Subsurface Geological Porous Media"

_remotesensing, doi:10.3390/rs15164094_

Round 1

Reviewer 1 Report

Comments for the manuscript entitled “A novel assessment of the atmospheric temperature role in radon (Rn-222) gas flow within subsurface geological porous media” by Benkovich et al., Remote Sensing. 

This is an interesting paper. However, the paper is not well organized and sometimes confused. I think the authors need to clarify these before it can be published.

 Major comments: 

11. This work is strictly not ‘remote sensing’ as it is normally meant but rather it is geophysics or solid Earth physics;

22. The terminology is also a bit confused. Such as “atmospheric temperature”.  It seems to me that it is the near-surface air temperature. If it is, please use the proper term and indicate what altitude the temperature is measured. In Figure 4, the authors said that the pink line is the temperature profile at 10 m. Is the temperature at 10 m the air temperature or soil/rock temperature?  It is also a bit strange to me that the temperature is the same for the whole day and almost same for winter and summer.

33. Figure title is too long. I think some explanation can be placed in the text. The font of paper is not consistent. There are no line numbers which make reviewers very difficult to write comments.

44. More important, it seems the writers have presented a number of correlated measurements which are not yet fully understood. The authors have measured near-surface atmospheric pressure and temperature using a standard weather station. For above surface sensing of the Rn, the role of air temperature and pressure in the process of exhalation may certainly be involved in measurements of Rn. In the well, the changing air temperature in the well may have played a similar role in both exhalation and measurement. But unfortunately, there are no measurements of air temperature in the well nor temperature (or Rn) in the rock/soil. The claim is that the measurements indicate Rn in the soil/rock pores. A question is therefore whether the correlated diurnal measurements are all or mostly interface effects?

The authors claim the diurnal and annual temperature variations in Rn measurements are indicative of an “… additional driving force at play” (Page 7 paragraph 3). They (in effect) imply there is a “downward temperature gradient” (Page 7, paragraph 3) in the rock/soil at work. This driving force would be the diurnal and annual input of solar radiation which creates the air temperature changes as well as a response in the Earth. The lags the authors mention would be governed by the heat equation to which may be added some decay energy. But there are no measurements that might help decide this. The near surface soil temperature is not monitored and no probes into the rock/soil are in place to monitor internal temperature. 

In short, it is not clear if the observations are not due to exhalation and the way the instruments measure Rn rather than a ‘driving force’ of the temperature gradient within the earth. There will certainly be an air temperature gradient in the well but it is not measured and there is no information about likely temperature gradients in the rock/soil.

The paper is, however, based on extensive analysis of real geophysical data and provides very interesting results. This would make it useful as a shorter letter or note to open the discussion of the causes to the wider audience. If the editors of Remote Sensing are able to include Geophysics in the journal’s scope it could well be a location for the letter or note.

Minor comments:

(1. In Equation 3, page 2,  Is P the atmospheric pressure?

(2.  Page 2. CRa is CRa?

(3.  Page 3, Figure 1 title, Zafrir et al., 2016 should use proper RS citation.

(4.  Page 11, equation (2) seems strange.

Author Response

Answer to Reviewer#1

Reviewer 2 Report

This paper describes the relationship of Semi-diurnal, diurnal, multiday, and seasonal Rn temporal variation to climatic temperature. The current written English seems good quality.

Author Response

Answer to Reviewer#2 is attached

Reviewer 3 Report

Dear Authors,

Reading this paper was a pleasure.

Following are suggestion how to improve:

1. Figure 1 (a,b,c) - I could not read text within images. I could be better resolution.

2. Note about most of the figures - almost all figures contains its explanation in form of title, which is good. But After that, there are sentences written explaining Figures, which are placed in Figure title. It should be places within paragraph instead. For example:

This is Figure 2 title:

"Figure 2. 2015-2018 Sde Eliezer measurements of Rn counts at 10, 40, 60 and 88 meters depth, and temperature, and pressure, at 15min resolution time. Between the dates 4/09/2017- 14/03/2018 there was equipment failure, and no measurements were generated. Detector gamma at 88m depth was added during December 2015. This detector was under, or partially- under the water table, except for June-December 2016 and hence the low counts. All radioactive datasets show a clear annual periodicity correlating positively with temperature and negatively with pres-sure. Conversion units between counts to Becquerel/m3 are: gamma sensors at depths 10 and 60m – 1.8 counts per 15min are equivalent to concentration of 1 Bq/m3; gamma sensors at depth 88m – 2.4 counts per 15min are equivalent to concentration of 1 Bq/m3; alpha particle detector - 0.08 counts per 15min are equivalent also to 1 Bq/m3."

Instead you could do it like this:

a) Figure title:

"Figure 2. 2015-2018 Sde Eliezer measurements of Rn counts at 10, 40, 60 and 88 meters depth, and temperature, and pressure, at 15min resolution time."

b) Text in paragraph explaining figure:

"       Between the dates 4/09/2017- 14/03/2018 there was equipment failure, and no measurements were generated. Detector gamma at 88m depth was added during December 2015. This detector was under, or partially- under the water table, except for June-December 2016 and hence the low counts. All radioactive datasets show a clear annual periodicity correlating positively with temperature and negatively with pres-sure. Conversion units between counts to Becquerel/m3 are: gamma sensors at depths 10 and 60m – 1.8 counts per 15min are equivalent to concentration of 1 Bq/m3; gamma sensors at depth 88m – 2.4 counts per 15min are equivalent to concentration of 1 Bq/m3; alpha particle detector - 0.08 counts per 15min are equivalent also to 1 Bq/m3."

Please use same idea for all figures with 'large' titles.

2. In Figure 2 there is: "there was equipment failure, and no measurements were generated" - could you write a bit more about equipment failure. Some readers could find that info usable when designing new one, since we all have problems with equipment.

3. You said that: "The analysis was conducted under Python environment using built-in and customized functions built upon libraries Datetime, NumPy, Pandas, and SciPy. libraries Matplotlib and Seaborn were used for visualization of the results." - could you write a bit more about specifics of analysis. A bit more about algorithm specifics.

4. Figure 7: 'in' is strike-through in text within following sentence: "overall model in (in bold black line)"

Thank you for effort.

Regards

Author Response

Answer to Reviewer#3 is attached 

Reviewer 4 Report

The paper "A Novel Assessment of the Atmospheric Temperature Role in Radon (Rn-222) Gas Flow within Subsurface Geological Porous Media" presents research to assess long-term Rn monitoring in depth during 2015-2018. The methods are precise, and the manuscript aligns with the journal's scope.

However, there are certain parts of the manuscript that require improvement. The relevance and importance of this research need to be clarified. Although the paper is novel, it lacks a clear presentation of its significance. I recommend that the authors highlight the results and how they are presented.

To address this, the introduction could conclude with a paragraph emphasizing the worldwide relevance of conducting these experiments and the significance of the acquired information. For instance, the first paragraph of the discussion could be a suitable place for such emphasis. This might require some restructuring of the manuscript.

Additionally, Figure 1 should include a geographical coordinate system to locate the study area accurately, as well as, a north arrow and a scale bar.

Furthermore, some graphs are presented in low quality, but they are crucial for understanding the manuscript. Therefore, it is essential to ensure all charts are of sufficient quality for readers to comprehend the research effectively.

Finally, I recommend increasing the discussion section with more worldwide references. Although well-written, this section needs improvement as this paper presents such results for the first time. Expanding the discussion with relevant references would enhance the overall impact of the research.

Author Response

Answer to Reviewer#4 is attached 

Round 2

Reviewer 1 Report

The authors have responded in detail to the comments although while some things have been clarified some are also still confusing.

It seems from the added material that the authors are proposing that there is a diurnal and seasonal, depth dependent variation in measurements of Rn due to or related to the temperature gradient in the rock/soil material. They provide evidence of this by relating the measurements to what they call (confusingly) ‘atmospheric surface temperature’.

The nomenclature is still confusing. It seems they must have an instrument near the well but at the surface where air temperature and pressure are measured. But this instrument is not shown in the Figures and no instruments or settings (e.g. height above the ground) are described for it. It seems different from the probes used to measure air temperature in the well. The text does not make sense to the reviewer unless there is such an instrument outside the well. If it is right that there is a weather station above ground: 

1 1. Please use ‘near surface air temperature and pressure’ to describe the measurements;

2 2. The title should be changed to something like “A novel assessment of the influence of surface heat flux on the flow of radon (Rn-222) gas within subsurface porous media”.

3 3. Surface temperature usually means radiometric surface temperature (skin temperature) measured by a thermal instrument with emissivity corrected. If emissivity is not corrected, then it is called the brightness temperature. Please check the term used in the paper.

Basically the authors are using the air temperature variation as a surrogate for the net radiation or the surface heat flux. It is not the same but they both have diurnal and seasonal variations. The site is dry and will likely have little vegetation so forcing will mostly be net radiation less sensible heat flux (wind speed). It would be possible for the authors to use a weather station and other information to estimate surface heat flux. The near surface ground heat flux could be more directly measured with buried heat flux plates. The actual ‘surface temperature’ (temperature of the surface) could be measured with radiometers. But the authors only have near surface air temperature. Still, they find very interesting results.

If there can be a clear statement of what is measured and why it can be taken as a surrogate for the time varying temperature gradient in the soil it will be useful to other researchers. With some knowledge of the density, heat conductivity and specific heat of the material the temperature gradient could be modelled with the heat equation based on the heat flux at the surface. But more information is needed for this. Perhaps the results here will lead to such work.

Author Response

Please see the attached response letter.

Reviewer 2 Report

All questions were carefully considered, proper corrections including adding explanations have been made in this version, and the novelty of this work has been clearly emphasized. Thus, it has met the publication scope and standards of Remote Sensing.

Author Response

(The authors gave the same response as above.)
